# Current State of Immunotherapy and Mechanisms of Immune Evasion in Ewing Sarcoma and Osteosarcoma

**DOI:** 10.3390/cancers15010272

**Published:** 2022-12-30

**Authors:** Valentina Evdokimova, Hendrik Gassmann, Laszlo Radvanyi, Stefan E. G. Burdach

**Affiliations:** 1Ontario Institute for Cancer Research, Toronto, ON M5G 0A3, Canada; 2Department of Pediatrics, Children’s Cancer Research Center, Kinderklinik München Schwabing, TUM School of Medicine, Technical University of Munich, 80804 Munich, Germany; 3Translational Pediatric Cancer Research, Institute of Pathology, Technical University of Munich, 81675 Munich, Germany; 4German Cancer Consortium (DKTK), German Cancer Research Center (DKFZ), Partner Site Munich, 81675 Munich, Germany

**Keywords:** Ewing sarcoma, osteosarcoma, immunotherapy, William Coley, tumor microenvironment, extracellular vesicles, exosome, immunosuppression, retrotransposon, human endogenous retrovirus

## Abstract

**Simple Summary:**

Pediatric sarcomas, including Ewing sarcoma and osteosarcoma, were first to be treated with an anticancer vaccine 100 years ago. This review moves on from a historical perspective to the current progress and challenges of immunotherapy in these immunologically cold bone and soft tissue sarcomas. We discuss mechanisms of immune escape and immunosuppression employed by these tumors, and the potential novel directions of the research and therapy. The intention of this review is to stimulate alternative concepts and treatment strategies.

**Abstract:**

We argue here that in many ways, Ewing sarcoma (EwS) is a unique tumor entity and yet, it shares many commonalities with other immunologically cold solid malignancies. From the historical perspective, EwS, osteosarcoma (OS) and other bone and soft-tissue sarcomas were the first types of tumors treated with the immunotherapy approach: more than 100 years ago American surgeon William B. Coley injected his patients with a mixture of heat-inactivated bacteria, achieving survival rates apparently higher than with surgery alone. In contrast to OS which exhibits recurrent somatic copy-number alterations, EwS possesses one of the lowest mutation rates among cancers, being driven by a single oncogenic fusion protein, most frequently EWS-FLI1. In spite these differences, both EwS and OS are allied with immune tolerance and low immunogenicity. We discuss here the potential mechanisms of immune escape in these tumors, including low representation of tumor-specific antigens, low expression levels of MHC-I antigen-presenting molecules, accumulation of immunosuppressive M2 macrophages and myeloid proinflammatory cells, and release of extracellular vesicles (EVs) which are capable of reprogramming host cells in the tumor microenvironment and systemic circulation. We also discuss the vulnerabilities of EwS and OS and potential novel strategies for their targeting.

## 1. Historical Perspectives and Modern Immunotherapy in Pediatric Sarcomas

### 1.1. William Bradley Coley (1862–1936): The First Focused Effort at Cancer Immunotherapy

Long before James Stephen Ewing, one of the leading US pathologists and the Medical Director of Memorial Hospital in New York, presented in 1920 the first case of a small round blue cell “endothelioma of bone” which since then carries his name [1], William B. Coley was treating hundreds of bone and soft-tissue cancer patients with a significant rate of success [2,3,4]. Being a renowned surgeon and the head of the Bone Tumor Service at the same hospital (nowadays Memorial Sloan Kettering Cancer Center), he pioneered the approach of stimulating the body’s “resisting powers”/immunity with the attenuated bacterial mixtures, paving the road to modern immunotherapy.

By 1896, Dr. Coley reported 160 cases of patients with inoperable malignant tumors treated with the mixture of heat-inactivated bacteria, the Gram-positive *S. pyogenes* (erysipelas) and the Gram-negative *Bacillus prodigiosus* (*Serratia marcescens*) [3], which became known as “Coley’s toxins”. Starting from 1899, these toxins were commercially produced by Parke Davis & Co and were widely available to the physicians in North America and Europe until 1951 [2]. Importantly, among a large variety of cancers treated with Coley’s toxins, the best response was achieved in patients with inoperable bone and soft tissue sarcomas, including EwS and OS, as well as in osteolytic types of rapidly growing malignant bone tumors (endothelioma, reticulum cell sarcoma and plasma cell myeloma), while carcinomas responded poorly to the treatment [5]. Coley himself concluded that the use of toxins should be limited to sarcomas [4].

Over the course of his 45-year-long career, Coley treated thousands of patients and published more than 150 monographs on the subject, and yet, his work came under scrutiny due to a number of reasons. These included variations in bacterial strain activity and lack of standard bioassays to test their efficacy prior to injecting into patients as well as a complexity and duration of their administration, making it difficult to standardize [5]. Furthermore, at the beginning of 20th century, radiation and then chemotherapy began to emerge as promising new cancer treatments (Figure 1). James Ewing, a big proponent of radiation therapy and Coley’s boss, forbid Coley from using his toxin inside the Memorial Hospital. The final blow was delivered in 1963, when the Food and Drug Administration (FDA) refused to recognize Coley’s toxin as a proven drug due to the lack of safety and efficacy data, despite over 70 years of its clinical use, hundreds of publications and, most importantly, remarkable improvements achieved by this treatment, including an independent controlled study in 1962 which showed a dramatic response in 20 of 93 cancer patients [6]. His legacy was saved by his children. Coley’s son Bradley succeeded him as the head of the Bone Tumor Service at the Memorial Hospital and advocated Coley’s toxin as adjunctive therapy for bone sarcomas after surgery to prevent metastasis [7]. His daughter, Helen Coley Nauts became one of the two founders of the Cancer Research Institute (CRI), which was established in 1953 and since then, is a leading institution in the field of cancer immunology and immunotherapy.

Ironically, the CRI-sponsored phase I clinical trial of Coley’s toxins conducted in 2012 in Germany revealed both promise and obstacles associated with its use in modern times. While the main objective of the trial was to establish safety and determine optimal dosing, one of 12 patients with metastatic bladder cancer had a clear clinical response, experiencing a 50% reduction in his cancer [8]. The team of researchers and clinicians led by Dr. Jäger faithfully recapitulated Coley’s treatment protocols, of which induction of fever is the key component of therapy. Patients received subcutaneous injections of Coley’s toxins twice a week, and the dose was escalated in each patient until a body temperature reached 38 °C to 39.5 °C, followed by four additional doses. Remarkably, coincident with the highest body temperature, ten of 12 patients showed a consistent increase in serum IL-6 levels, and some (including a patient with metastatic bladder cancer) also manifested an increase in TNF-α, IFN-γ and IL-1β. The team concluded that fever-induced massive release of immunoregulatory cytokines may play an important role in tumor regression and suggested further exploration of Coley’s vaccine as a potent immune modulator [8]. However, further testing of Coley’s vaccine is under scrutiny again. The major obstacles include the cost and complexities associated with manufacturing a bacteria-derived product in accordance with Good Clinical Practice guidelines. Most importantly, no modern hospital institutional review board will permit maintaining fever as one of the study objectives. Nevertheless, growing number of pre-clinical and clinical studies into the use of bacteria to combat cancer provides additional rationale for implementing basic principles established by Dr. William Coley 100 years ago into the clinical practice [9].

### 1.2. Conventional and Targeted Therapeutics in Treating EwS and OS

Nowadays, high-risk childhood bone cancers, including EwS and OS, range among the leading causes of cancer-related deaths in children. Despite a significant increase in the 5-year survival rates among EwS patients with localized disease from less than 10% in the pre-chemotherapy era to about 75% currently, outcomes have not significantly improved since late 1990s (Figure 1). Likewise, combination chemotherapy introduced in the 1970s for treatment of OS increased overall survival rates for patients with localized disease from 20% to approximately 70% today, however patients with metastatic or recurrent disease fare poorly with only 20% 5-year survival rates [10,11]. The standard protocols for both EwS and OS patients rely upon extensive surgical resections, high-dose multimodal chemotherapy and radiation [12,13,14,15], putting young survivors at risk of developing long-term disabilities and therapy-induced secondary cancers [16]. Furthermore, the relapse rates remain high and less than 30% of patients with metastasis survive for 5 years [12,13,14], urging the need for developing tailored therapy approaches, especially for these high-risk groups of patients.

In the past 20 years, multiple new-generation inhibitors of oncogenic driver proteins and pathways have been tested in clinical trials to enhance efficacy and overcome resistance to conventional therapies. Due to different reasons, targeting driver mutations in EwS and OS proved to be challenging. Unlike OS which has no clear driver and is uniformly characterized by high genomic instability and chromothripsis due to inactivating mutations in various DNA repair genes, EwS possess relatively stable genome with no recurrent mutations and gene amplifications [17]. Instead, it is driven by a single well-defined driver, the EWS-FLI1 fusion protein, which is found in ~85% of EwS cases [12,13]. Yet, EWS-FLI1 lacks enzymatic activity, and its direct chemical targeting was unsuccessful so far. Indirect EWS-FLI1 inhibition includes targeting downstream interacting proteins (e.g., RNA helicase A using YK-4-279, TK216 inhibitors), EWS-FLI1-driven epigenetic and transcriptional reprograming (e.g., using trabectedin and mithramycin), activated pathways (most notably, IGF1R and MEK), epigenetic modifiers (e.g., deacetylase, LSD, BRD4 and EZH2 inhibitors), anti-angiogenic tyrosine kinase inhibitors, and various combinatorial treatments; an overview of the past and current clinical trials is provided in [15,18,19]. Remarkably, despite their distinct biological and clinical properties, both EwS and OS exhibit gene expression profiles characteristic of BRCA1/2 mutant tumors (so-called “BRCAness”) [20,21], and thus expected to be susceptible to the poly(ADP)-ribose polymerase (PARP) inhibitors olaparib and talazoparib. Unfortunately, adding these newer agents to the existing treatment protocols has not improved outcomes, with the acquired and intrinsic resistance to therapy impeding further progress [14,15]. It becomes increasingly evident that without re-activating the immune system and re-thinking the designs of clinical trials, conventional and targeted therapeutics may not substantially improve outcomes in EwS and OS, where standard-of-care has not been significantly changed in decades [10,11,22], as well as in the majority of other solid malignancies [23].

### 1.3. Immunotherapy of Pediatric Bone Cancers

Complete or partial tumor regression and prolonged remission in some sarcoma patients treated with Coley’s vaccine provided one of the first clues that activating natural immunity may have anti-tumorigenic effects. Modern immunotherapies are aimed at building strong antitumor immunity by re-activating and amplifying pre-existing antitumor responses and inducing new responses against otherwise cryptic antigens (e.g., with checkpoint inhibitors, oncolytic viruses, and cancer vaccines) or by adoptive cell therapies, including engineered CAR T cells, TCR transgenic T cells or tumor-infiltrating lymphocytes (TIL) (Table 1). Their implementation in the past decade offered exceptional clinical benefits for patients with certain types of “immunogenic” cancers but even so there are unprecedented challenges, ranging from developing pre-clinical models, defining dominant drivers of anticancer immunity and immune escape to the off-target toxicities and the need to identify optimal combinations for any given patient [24,25]. The majority of solid tumors remain immune “cold” and at the outset unable to generate an appreciable de novo immune response that can be capitalized on by immunotherapy, owing to low mutational burden, lack of MHC expression and other factors.

For example, targeting overexpressed surface molecules, including HER2 [26], B7-H3 [27] and the gangliosides and disialogangliosides (GD2 and GD3) [28,29,30] using antibody-based therapy and CAR T cells is a promising therapeutic strategy in EwS and OS. Preclinical and clinical studies have indicated that targeting GD2 generates robust anti-tumor response in EwS and in some other tumor entities including neuroblastoma [30,31], however off-target neurotoxicity can occur [32]. The ongoing clinical trials will test its efficacy in patients with EwS and OS (NCT02107963, NCT04539366).

Similar to the majority of solid tumors in adults [23,33], the above immunotherapy strategies showed limited efficacy in pediatric cancers, including EwS and OS [19,22,34,35]. As discussed below, this is mainly because of the lack of tumor-specific antigens (TSAs) and neoantigens [36,37,38], low expression of the classical human major histocompatibility complex class I (MHC-I) and thus impaired MHC-peptide presentation [39], upregulation of checkpoints and the immunosuppressive tumor microenvironment (TME), which is mostly populated by pro-tumorigenic M2 macrophages. One of the emerging immune escape mechanisms in cancer also involves an increased secretion of TSAs, TSA-MHC complexes, oncogenic proteins and RNAs in extracellular vesicles (EVs) [40,41,42].

**Table 1 cancers-15-00272-t001:** Immunotherapy of pediatric bone cancers.

Approach	Goal	Target	Therapeutic Agent	Major Obstacles	Refs
Immune checkpoint inhibitors	Reactivating and amplifying preexisting antitumor immunity	PD-1	Nivolumab/OPDIVO^®^ Pembrolizumab/ Keytruda^®^	Low expression, low mutational burden, Immunological cold TME	[19,43,44,45,46,47,48,49,50,51,52,53,54,55]
		PD-L1	Atezolizumab/Tecentriq^®^		
		CTLA-4	IpilimumabYERVOY^®^		
Tumor specific antigens (TSAs)	Direct tumor targeting	GD2	Dinutuximab/Unituxin^®^ anti-GD2 CAR-T cells anti-GD2 CAR-engineered NK cells	Variable expression in EwS and OS Upregulation of HLA-G checkpoint	[28,29,30,56]
		IGF1R	Ganitumab, Dalotuzumab	Activation of compensatory mechanisms, Toxicity	[57]
		HER2	Trastuzumab/Herceptin^®^	Not expressed in EwS, no clinical benefit for OS	[18,45]
		B7-H3	Anti-B7-H3 CAR T cells		[27]
Antitumor vaccines	Direct tumor targeting	Tumor TSAs or proteins	Dendritic cell vaccine	Need for autologous DCs, Labor-intensive and costly cell isolation	[58]
	Activation of DC responses	Multiple tumor antigens	Attenuated tumor cells, could be pulsed with GM-CSF, IL-2 or IL-7 or siRNAs	Immunosuppressive TME, low tumor immunogenicity	[22]
Oncolytic viruses	Increase tumor immunogenicity Induce immunogenic cell death	Tumor	Vaccinia virus/Pexa-Vec Reovirus/Reolysin HSV-1/HSV1716 Adenovirus X-Vir	Antiviral immunity, Low delivery efficacy, Immunosuppression, T cell exhaustion	[34,59,60,61,62]
Targeting immunosuppressive TME	Macrophage activation	TME	L-MTP-PE/Mifamurtide, BCG, Coley’s toxins, oncolytic viruses		[63,64]
	Macrophage polarization	TME	All-trans retinoic acid (ATRA)	Low delivery efficacy	[65]
	Macrophage/MDSC depletion	TME	All-trans retinoic acid (ATRA) Trabectedin	Toxicity	[66,67,68,69]

## 2. Mechanisms of Immune Escape

A better understanding of immune evasion mechanisms employed by tumor to avoid recognition and killing by the immune system is a key to successful therapy aimed at combating cancer with minimally invasive and non-toxic modalities. To search for cure and not just for 5-year overall survival for pediatric patients, it is imperative to establish critical aspects of the interplay between the tumor and the immune system. Several key points are discussed below.

### 2.1. Lack of Tumor-Specific Antigens (TSAs)

Targeted therapeutics and immunotherapy strategies are critically dependent on identification of druggable TSAs and neoantigens. To achieve maximal clinical efficacy and minimal toxicity, the ideal target should be immunogenic, highly expressed and presented on the surface of the majority of tumor but not normal host cells, and play a role in tumorigenesis. A very few antigens in general (and none of them in pediatric sarcomas) meet such criteria. In contrast to adult cancers, pediatric tumors exhibit very low mutation rates and, consequently, much fewer TSAs and tumor neoantigens [22,34,35]. Even OS, which has relatively high level of copy-number variations and gene deletions [70], exhibits on average about 7 neoepitopes per tumor and only 2 of them are predicted to be expressed, while EwS has none [71]. With regard to cell surface proteins, the best studied targets include B7-H3, GD2, IGF1R (which are shared between OS and EwS) as well as HER2 (for OS) and CD99, endosialin/CD248, TRAIL-R and STEAP-1 (for EwS) [18,34,45]. These are being targeted using CAR T cell approaches, bispecific T-cell engagers and antibody-drug conjugates. While multiple pre-clinical development programs and clinical attempts are still ongoing [15,18,19], none of them has shown significant clinical benefit so far, in part due to toxicities associated with their expression in other tissues. Targeting intracellular TSAs is also problematic, mainly because of low expression of MHC-I molecules that are required for the intracellular peptide presentation to the effector CD8^+^ T cells [39], see also Section 2.2. However, class I expression may pertain to advanced EwS and experimental approaches have been established to induce it [72,73,74].

One of the potential solutions to the lack of conventional TSAs in pediatric sarcomas may be hidden in noncoding regions of the genome, including introns, alternative splicing variants, gene fusions, endogenous retroelements and other unannotated open reading frames [75]. According to recent studies, noncoding regions could be the main source of targetable TSAs in human malignancies [76,77,78,79]. Activation of these regions mainly occurs due to demethylation of the genome and other epigenetic mechanisms that are highly dysregulated in tumor cells, rising a possibility that the resulting TSAs may be widely shared between different tumor types and absent from normal tissues [80,81]. In addition, pediatric sarcomas may also express unique neoantigens from noncoding regions, given that at least a third of them carry recurrent chromosomal translocations and express characteristic fusion proteins which act as transcriptional and epigenetic regulators [13,15,82]. For instance, in EwS, epigenetic changes are driven by EWS-FLI1, through its ability to bind to GGAA microsatellite sequences [13,14]. A recent report provided the first evidence that EWS-FLI1 and potentially other EWS fusions can drive transcription, processing and translation of neopeptides from the silent genomic regions including GGAA microsatellite repeats [83], although their presentation on MHC-I, immunogenicity, TCR addressability and druggability remains to be studied.

### 2.2. Low Expression of MHC-I and Upregulation of Immune Checkpoints

One of the mechanisms whereby pediatric sarcomas escape T cell-mediated immunosurveillance is impaired expression of MHC class I/Human Leukocyte Antigen (HLA) class I antigens [39]. About 48–79% of primary EwS and the majority of metastatic lesions, especially pulmonary metastasis, exhibit low-to-absent MHC class I and II expression [84,85]. In EwS, low levels of MHC-I correlate with reduced CD8^+^ T cell infiltration and survival [85]. Likewise, ~25–52% of primary and 44–88% of metastatic OS manifest complete loss or downregulation of MHC-I, being strongly associated with decreased survival [46,86,87].

Upregulating MHC-I expression in pediatric sarcomas may thus be a promising strategy to activate CD8^+^ T cell-mediated antitumor responses [88]. This can be achieved by stimulating proinflammatory pathways, such as TNF-TNF receptor-NFκB, type I IFNs-IFNAR1/2-STAT1/2/3 or type II IFN-IFNGR-STAT1 [89]. For example, MHC-I expression in EwS cell lines is induced by IFNγ or mediators of dendritic cell maturation, including TNF [84,90,91]. In particular, treatment with GM-CSF, IL-4, TNF, IL-6, IL-1β and PGE_2_ upregulated MHC-I, ICAM-1 and CD83, and improved recognition of EwS cell lines by TSA-specific TCR transgenic T cells in vitro [74]. In clinical settings, certain treatments including irradiation and hyperthermia may induce TNF secretion and MHC-I expression and may thus render EwS cells more susceptible to MHC-I-dependent immunotherapies [74,92,93]. The MHC-I presentation on tumor cells can also be induced by adenovirus-based oncolytic virotherapy. Oncolytic adenoviruses selectively replicate in tumor cells, where they induce the cGAS-STING pathway [94], and promote MHC-I expression. One example is the YB-1-targeting adenovirus XVir-N-31, which demonstrated substantial antitumor activity in a murine EwS xenograft model in combination with CDK4/6 inhibitors [62]. It is thus likely that combinatorial treatments would be necessary to increase MHC-I expression and CD8^+^ T cell-mediated antitumor immunity.

Concurrent with reduced MHC-I expression, upregulation of immunosuppressive receptors and checkpoints may contribute to the immune escape in pediatric sarcomas. For instance, HLA-G and HLA-E, the non-classical MHC-I molecules implicated in the protective maternal-fetal barrier in the placenta [95], are highly upregulated on tumor and myeloid cells in the EwS TME. HLA-G is expressed in ~34% of EwS biopsies, and can be further induced by proinflammatory signaling, including IFNγ and GD2-specific CAR-engineered NK cells [56,91,96]. When expressed on tumor cells, HLA-G and HLA-E were shown to interact with inhibitory receptors expressed on T cells and NK cells, negatively affecting cytotoxic functions of both CD8^+^ T cells and NK cells [97]. Yet, in experimental in vitro model, ectopic expression of HLA-G by myeloid THP-1 cells but not by EwS tumor cells impeded functionality of CAR T cells, suggesting that immunosuppressive effects of HLA-G could be mediated by myeloid cells in the TME [96]. However, expression of various HLA-G isoforms and lack of specific antibodies are currently hampering the development of HLA-G targeting approaches in EwS and other tumors [97].

On a similar note, therapeutic targeting of PD-L1 and PD-1 immune checkpoints, which are expressed in ~20% of pediatric sarcoma patients in EwS and OS [46,47,48,49,50,51,52], has not shown clinical efficacy [43,53,54]. In line with low expression of immune checkpoints on EwS and OS tumor cells, only 35% of EwS- and OS-infiltrating immune cells express PD-L1 [47], whereby expression of PD-L1 or PD-1 on T cells is rare for the most part in EwS and OS [50,55,98,99], and is predominantly observed on macrophages [54]. Yet, ~10% of post-treatment OS tumors score in the top quartile of immune infiltration, which is comparable to other strongly immune-infiltrated malignancies, including lung cancer and renal clear cell carcinoma [100]. The respective groups of OS patients may potentially benefit from the immune checkpoint blockade.

### 2.3. Immunosuppressive TME in EwS and OS

#### 2.3.1. Improving CD8^+^ T Cell Infiltration and Antitumor Activity

Only 12–38% of EwS and ~52–68% OS tumors are infiltrated by cytotoxic CD8^+^ T cells [48,55,91]. Poor CD8^+^ T cell infiltration is a negative prognostic marker associated with metastatic progression and worse outcomes [85,99,101,102,103,104]. Key chemotactic mechanisms for the recruitment of TILs and activation of antitumor immune responses are the TME-derived C-X-C Motif Chemokine Ligand 9/10 (CXCL9/10) or stromal-derived Chemokine (C-C motif) ligand 5 (CCL5) and their respective receptors C-X-C Motif Chemokine Receptor 3 (CXCR3) and CCR5 [105,106]. In EwS, increased expression of CXCL9/10 and CCL5 correlates with infiltration of CD8^+^ CXCR3^+^/CXCR5^+^ T cells [103]. However, in recurrent pediatric sarcomas, EwS-infiltrating macrophages express lower levels of CXCL9/10 compared to OS [107]. Inducing CXCL10 expression in the EwS TME in order to enhance T cell infiltration may thus be a promising therapeutic strategy. This can be achieved by intratumoral injection of attenuated pathogens or oncolytic viruses [108], especially those harboring CXCL10 transgene [109], as well as by using dipeptidylpeptidase 4 (DPP4) inhibitors to stabilize CXCL10 [110,111].

In spite of higher proportion of TILs in the OS TME compared to EwS, they exhibit terminally exhausted phenotypes, including expression of co-inhibitory receptors TIGIT, LAG3, PD-1 and TIM3 [107]. Apart from the PD-1/PD-L1 axis, their blockade in OS may thus enhance TILs cytolytic activity. This may be especially relevant in OS with pulmonary metastases, which show increased T cell infiltration at the interface between the adjacent healthy tissue and tumor stroma [100,112]. These interfaces are enriched with activated exhausted CD8^+^ T cells positive for PD-1, LAG3 and IFNγ and with myeloid cells expressing M-MDSC and DC signatures. The core of pulmonary metastases is devoid of immune infiltrates, suggesting that myeloid cells may exclude TILs [112].

In contrast to OS, circulating lymphocytes in EwS exhibit mixed gene expression profiles associated with effector responses (granzyme A and B, and perforin) and intermediate exhausted phenotypes (TIM-3) [107]. This is partly in line with previous results observing increased frequencies of circulating PD-1^+^CD4^+^ and CD8^+^ T cells [113], however immune checkpoint blockade failed to show clinical response, as discussed above.

#### 2.3.2. Targeting Tumor-Associated Macrophages

The most abundant immune cells in the TME of EwS and OS are tumor-associated macrophages (TAMs) which exhibit immunosuppressive M2 signatures [104,114,115,116]. Based on recently published transcriptomic analysis, these M2 macrophages may be phenotypically and functionally distinct in EwS and OS [107]. In line with this, TAM infiltration in EwS was indicative of poorer survival [104,115,117], while opposite observations were made in OS, where infiltration with CD14^+^/CD163^+^ myeloid cells and M1/M2 macrophages correlated with improved outcomes [67,99,118]. However, infiltration with CD68^+^ macrophages was associated with worse survival in OS [49], suggesting the existence of different TAM populations with opposite activities. Higher density of CD68^+^ and CD163^+^ macrophages in OS (the CD68^+^ to TIL ratio is 5.9, compared to 2.5 in EwS) may contribute to OS aggressiveness [119].

Chemotactic signals from the TME recruit monocytes from the bone marrow into the tumor stroma [120], where they polarize into TAMs and pro-tumorigenic M2 macrophages [121] (Figure 2). Signaling in the TME promotes sarcoma progression by inducing angiogenesis [115,122,123], migration [124], extravasation [125] and chemotherapy resistance [126]. TAMs in pediatric bone sarcomas release pro-inflammatory cytokines [115], prevent T cells from entering the tumor core [112] and impede the activation and degranulation of T cells [127]. Targeting TAMs may therefore be a promising therapeutic strategy, which is currently being investigated in multiple pre-clinical studies and clinical trials. This involves the following directions:Macrophage activation using liposome-encapsulated muramyl tripeptide phosphatidyl ethanolamine (L-MTP-PE or mifamurtide), a constituent of the Mycobacterium cell wall originally purified from the attenuated *Mycoblasma bovis*, also known as Bacille Calmette-Guerin (BCG). BCG vaccine is currently used for treatment of certain types of cancer and may have a mechanism of action similar to Coley’s toxins. The synthetic L-MTP-PE was shown to stimulate macrophages and to improve survival in OS [63,64,128,129]. Another approach to activate TAMs are oncolytic viruses, which are capable of inducing immunogenic tumor cell death (ICD). This is accompanied by release of TSAs and danger- and pathogen-associated molecular patterns (DAMPs and PAMPs), switching TAMs to antitumorigenic M1 macrophages [130].Blocking the immunosuppressive M2 polarization and depleting myeloid-derived suppressor cells (MDSCs) using all-trans retinoic acid (ATRA). Treatment with ATRA reduced metastasis in an OS mouse model [65] and improved the efficacy of CAR T cells against pediatric sarcomas in vivo [66].Depleting TAMs and MDSCs using chemotherapeutic agents. For example, trabectedin, which is a natural product from sea squirt shown to inhibit transcription factor bindings such as FUS-CHOP in myxoid liposarcoma or EWS-FLI1 in EwS [131], enhanced CD3^+^ T cell infiltration in OS and other cancers as well as oncolytic virotherapy against EwS xenograft in a mouse model [67,68,69]. Trabectedin is currently tested in a clinical trial for EwS (NCT04067115).Other therapeutic options include block of recruitment and reprogramming metabolic switches.

**Figure 2 cancers-15-00272-f002:**
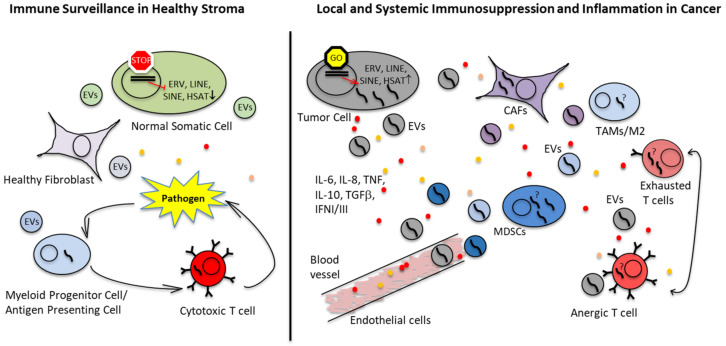
Immunosuppressive tumor environment and immune evasion mechanisms in EwS and OS. Oncogenic drivers and epigenetic changes in EwS, OS and other cancers activate proinflammatory pathways and increase secretion of cytokines, chemokines, growth factors and EVs. Proinflammatory microenvironment attracts resident tissue macrophages and blood-circulating monocytes, while skewing their differentiation and promoting accumulation of M2 macrophages and immature myeloid cells with tolerogenic properties. Together with cancer-associated fibroblasts (CAFs), these cells (TAMs/M2 and MDSCs) constitute a majority in the TME. They fail to efficiently activate T cells and, instead, may induce T cell anergy and exhaustion. Additional contributing mechanisms may include re-expression of retroelements (LINEs and SINEs) and endogenous retroviruses including HERV-K in tumor cells. Dissemination of these virus-like RNAs in tumor EVs and their potential uptake by immune cells and CAFs may induce innate immune responses in these cells, leading to chronic inflammation. Tumor EVs bearing TSAs and pre-formed TSA-MHC complexes may also function as a decoy to divert antitumor immunity from cancer cells, especially when taken up by bystander host cells.

Therapeutic approaches directed at targeting TAMs and summary of clinical trials in sarcomas are discussed in detail elsewhere [132,133,134].

### 2.4. Immunogenicity and Response to Immunotherapy of EwS and OS in the Context of Bone and Soft Tissue Sarcomas

Bone and soft tissue sarcomas (STS) of children and adults comprise a heterogenous group of tumors with distinct biological properties, albeit they all share non-immunogenic properties and non-responsiveness to immunotherapy [22,43,135,136,137]. The most common bone tumors include EwS, OS and chondrosarcoma, while fibrosarcoma, gastrointestinal stromal tumors (GIST), leiomyosarcoma, liposarcoma, rhabdomyosarcoma (RMS), undifferentiated pleomorphic sarcoma and synovial sarcoma are the most frequent STS [138].

Immune infiltrates are heterogenous between sarcoma entities and age-dependent [139]. Sarcomas driven by mutations and copy number alteration tend to be T cell-inflamed, while translocation-driven sarcomas are immunologically cold [101,140,141]. Remarkably, mutation rates among sarcomas are low compared to other tumor entities [142], with pediatric sarcomas exhibiting the lowest numbers of mutations per Mb (EwS 0.24; OS 0.38 and RMS 0.33, the most frequent pediatric STS, compared to adult STS 1.06) [141,143]. Expression of immune checkpoints and clinical response to immune checkpoint blockade in sarcomas is dynamic, variable, and dependent on the histologic subtype. PD-L1 expression is sparse on the pediatric sarcomas EwS, OS and RMS, similar to the majority of adult STS (~20% PD-L1 expression) [144,145,146]. Consistent with this, the majority of pediatric and adult sarcomas do not respond to immune checkpoint blockade and predictive markers for responsiveness to ICI are yet to be determined [54,147]. The exceptions are the undifferentiated pleomorphic sarcoma and alveolar soft-part sarcoma, which are inflamed tumors with higher mutational burden [43,142], positive for PD-L1 in up to 40% of cases and partially responsive to immune checkpoint blockade [145,148,149]. Specific immunotherapeutic approaches in sarcomas have been reviewed elsewhere [22,26,137,150,151].

### 2.5. Extracellular Vesicles (EVs) as Means of Immune Escape

Communications between tumor and host cells in local and distant tumor sites are mediated by diffusible molecules such as cytokines, chemokines and lipids as well as to a large extent by extracellular vesicles (EVs) that create permissive environment for tumorigenic progression (Figure 2). EVs do so by transferring nucleic acids, proteins, lipids and various metabolites from tumor to various host cells, and vice versa [152,153,154]. As such, the EV cargo reflects the cell of origin and its physiological conditions, representing an important source of cancer-associated biomarkers. Most importantly, the EV cargo is encapsulated into lipid bilayer membrane and thus protected from degradation. When taken up by bystander normal and malignant cells, the EV cargo and is capable of functionally reprogramming the acceptor cells.

The EVs are comprised of highly heterogeneous populations of vesicles, whose secretion and composition are influenced by environmental conditions and tissue homeostasis. Most studies are focused on the nanosized vesicles (30–200 nm) originating from endosomal compartments (exosomes) or plasma membrane (ectosomes), which are believed to be important players in extracellular communications in healthy and diseased states [154,155,156]. The original hypothesis proposed in early 1980s implicated exosomes (EVs) as garbage bags for removal of unwanted proteins or harmful metabolites from the cells [157,158]. Indeed, secretion of cellular waste in EVs (or in specialized EV subsets) may be important for maintaining cellular homeostasis in normal and cancer cells. Recent findings have indicated that secretion in exosomes is essential for removal of damaged DNA and that blocking exosomal pathways provokes innate immune responses and induces senescence-like phenotype or apoptosis in normal cells due to accumulation of nuclear DNA in the cytosol [159]. Packaging in EVs is also required for expulsion of chemotherapy drugs and cellular toxins [160,161]. The EV-mediated waste management may be especially important for cancer cells, given their high proliferation and metabolic rates, and deficiencies in DNA repair pathways.

As discussed in numerous comprehensive reviews, tumor-derived EVs influence all major hallmarks of cancer, including immune evasion, tumor-promoted inflammation, angiogenesis, metabolic and epigenetic reprograming of the recipient cells, extracellular matrix remodeling, cancer metastasis and drug resistance [40,153,162]. In bone sarcomas and in cancers that preferentially metastasize to bones (such as prostate and breast carcinomas), EVs secreted by tumor cells are also capable of interfering with osteogenesis to promote tumor-supporting microenvironment inside the bone [163,164,165,166]. Particularly, OS-derived EVs enhanced angiogenic activities of endothelial cells, triggered macrophage dedifferentiation and increased a number of osteoclast-like cells and cancer-associated fibroblasts (CAFs) in local TME and metastatic sites [167,168]. Using murine NIH3T3 fibroblasts which are more susceptible to oncogenic transformation, it was also shown that OS EVs may induce tumor-like phenotype in non-transformed cells [169]. Moreover, OS EVs promoted epigenetic changes in mesenchymal stem cells (MSCs) but not in pre-osteoblasts, indicating that MSCs are highly susceptible to the EV-mediated epigenetic transformation [170]. Given that MSCs of osteogenic lineage are believed to be potential cells of origin for OS [171], their reprogramming by OS EVs can be an early event during OS development. Other mechanisms may involve a membrane-associated form of TGFβ which is transported in OS EVs to MSCs, leading to enhanced production of the proinflammatory cytokine IL-6 [172]. Likewise, IL-6 secreted by stromal cells in EwS TME was shown to contribute to EwS progression by protecting from apoptosis and promoting migration [173]. Using IL6- and TGFβ-blocking agents may thus be a viable therapeutic option for OS and EwS patients [172].

Lack of TSAs and low expression of MHC molecules have been described in previous sections as one of the major impediments for therapeutic targeting of EwS and OS cells. The available evidence suggests that their release in EVs could be one of the mechanisms employed by tumor cells to eliminate their specific antigens and MHCs and to reduce their recognition by cytotoxic T cells. Indeed, presence of tumor-derived MHCs and antigens (including pre-formed functional TSA-MHC complexes) in EVs is a well-documented phenomenon [40,41,42], albeit its role in EwS and OS remains to be elucidated. Dissemination of tumor EVs harboring TSAs and their subsequent acquisition and cross-presentation by bystander immune and non-immune cells may also act as a decoy to divert antitumor immunity from cancer cells. Remarkably, EVs carrying MHC-peptide complexes can directly activate cognate receptors on T cells, but the efficacy of antigen presentation is increased when EVs are attached to the surface of mature DCs [41,42]. However, this mechanism known as cross-dressing [174] may be compromised in tumor settings, given accumulation of MDSCs and immature DCs.

EwS EVs may be directly involved in generation of immature proinflammatory myeloid cells in local TME and systemic circulation. We showed that EwS EVs induced secretion of IL-6, IL-8 and tumor necrosis factor (TNF) by primary CD33^+^ myeloid cells and CD14^+^ monocytes, and inhibited their maturation into antigen-presenting DCs [175]. In particular, CD14^+^ cells differentiated in the presence of EwS EVs exhibited a semi-mature phenotype and immunosuppressive activity, including reduced expression of co-stimulatory molecules CD80, CD86 and HLA-DR, activation of the innate immune response gene expression programs, and the ability to interfere with activation of CD4^+^ and CD8^+^ T cells. Therefore, EwS EVs may contribute to systemic inflammation and immunosuppression by skewing differentiation and maturation of blood-circulating and tumor-infiltrating myeloid cells.

Mechanistically, induction of immunosuppressive myeloid cells is primarily mediated by various protein and RNA constituents present in tumor EVs [176]. EwS EVs carry multiple mRNAs encoding oncogenic drivers, including EWS-FLI1, EZH2 and stem cell-associated proteins [177,178], some of which can be transferred to the neighboring mesenchymal stem cells [179]. Whether or not EV-derived RNAs are actually capable of driving a sustainable protein expression in the recipient cells is an open question, given that the majority of these RNAs, including mRNAs and microRNAs, are severely fragmented and present in less than one copy per EV [180,181]. Additionally, given that the RNA lifetime is relatively short, massive production of RNA-containing EVs and their highly selective uptake might be required to reprogram the recipient cells by a particular RNA in EVs [180].

Of further consideration, our recent whole transcriptome analysis of EVs isolated from plasma of EwS patients and cell lines showed that the vast majority of RNAs in these EVs (up to 70–90%) are derived from satellite repeats, endogenous retroelements and retroviruses (ERVs), including *HSAT2*, *LINEs*, *LTR*/*ERVs*, *SINE*/*Alu*, and *7SL RNA* [182]. Similar results were reported for EVs from low-passage brain cancer cell lines [183] and patient-derived glioma stem cell-like cultures [180], as well as from co-cultures of breast cancer cells with stromal fibroblasts [184]. Compared to the respective parental cells, these EV preparations exhibited a significant enrichment with *ERV, LINE, SINE* and other repeat RNAs. Transcriptional activation of the respective heterochromatic genomic regions could be attributed to demethylation of the genome and other epigenetic changes characteristic of many if not all human malignancies [80,81]. In turn, their release in EVs could be a protective mechanism directed at preventing activation of innate immune responses in tumor cells, given highly immunogenic virus-like features of these transcripts. Moreover, we [182] and others [183] provided preliminary experimental evidence that at least some of these RNAs may be transferred in tumor EVs to normal cells, essentially mimicking viral infection (Figure 2). In contrast to other RNAs in EVs, retroelement-derived transcripts including *LINE-1* and *HERV-K* may retain some coding and replicative potential and may potentially be able to propagate in the “infected” cells via reverse transcription mechanisms [185,186]. Further investigation into these mechanisms may provide unique insights into cancer-associated inflammation, evasion of antitumor immunity and immunosuppression, and open novel directions for therapeutic targeting.

## 3. Conclusions and Perspectives

The development of efficacious immunotherapies in immune-inert pediatric sarcomas requires addressing both the tumor cell itself and the TME. Immunogenicity of the tumor cell can be increased by upregulating MHCs and/or immune stimulatory molecules such as CD83 and ICAM-1 on the tumor cell surface, or by increasing its sensitivity to immune checkpoint inhibitors. A combination of these approaches is warranted. In turn, immunogenicity of the tumor environment can be enhanced by altering macrophage differentiation and polarization or by administering activating cytokines. Additional tumor microenvironment-directed approaches could be designed to interfere with the immunosuppressive mechanisms active in the immunological deserts, e.g., blocking immunosuppressive EVs or immunosuppressive metabolic mechanisms, or probably both. This can be achieved by engineering bifunctional TCR or CAR transgenic T cells that could simultaneously manipulate the TME and target tumor-specific cell surface antigens. Moreover, epigenetic activation of gene expression from non-coding sequences may provide targetable neo-epitopes even in immune inert malignancies. Finally, expression of immunogenic neo-epitopes from non-coding sequences can be combined with repression of non-coding sequences associated with immune suppression or tolerance. Multifunctional T cell engineering can be envisioned, where transgenic TCR or CAR T, or NK cells recognizing tumor cells are designed to co-express immune stimulatory and TME manipulating gene products, such as those affecting macrophage polarization and/or molecules repressing EVs. Lastly, a better understanding of the role of sarcoma EVs in mediating immune dysfunction and ways of alleviating this are needed. We also need to establish whether EV-induced immune disruption (e.g., via releasing retroelement RNAs and inducing systemic inflammation) is a more widespread phenomenon across other cancers.

## Figures and Tables

**Figure 1 cancers-15-00272-f001:**
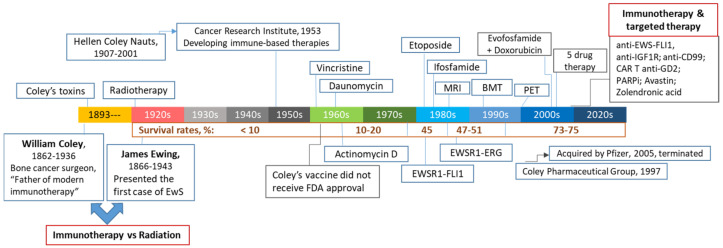
The 100-year-old battlefield between immunotherapy and radiotherapy in treating patients with bone and soft-tissues sarcomas. Milestones and major events are documented in a chronological order.

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
