# Peer review of "Current State of Immunotherapy and Mechanisms of Immune Evasion in Ewing Sarcoma and Osteosarcoma"

_cancers, 2022, doi:10.3390/cancers15010272_

Round 1

Reviewer 1 Report

In this manuscript the authors present a review of the literature on the potential novel immunotherapy strategies and on the mechanisms of immune escape of two subtypes of sarcoma : osteosarcoma and Ewing sarcoma. These two childhood bone cancers are indeed among the leading causes of cancer-related deaths in children. No significant improvement for outcomes has been made for the patients in the last two decades and there is an urgent need to find innovative therapies targeting these tumors and those indeed include immunotherapy strategies.

This is an interesting review, well-written, clearly presented and well-illustrated which provides useful information.

I only have the following minor comments :

-          Although the focus of the review is on osteosarcoma and Ewing sarcoma, since sarcoma represent a highly complex and heterogeneous group of tumors, it would be useful to have a short paragraph replacing these two childhood bone sarcomas in the context of the other sarcomas regarding immunogenicity and immunotherapy approaches.

-          Trabectedin is mispelled : Trabectidin instead of Trabectedin.

-          The disialoganglioside GD2 appears as a promising immunotherapeutic target for osteosarcoma and Ewing sarcoma, it is highly expressed especially in Ewing sarcoma. The authors might therefore develop the potential of this strategy notably the GD2-directed CAR-T cell approach.

Reviewer 2 Report

In this review, firstly, historical perspectives and modern immunotherapy in pediatric sarcomas, especially immunotherapy were introduced. Moreover, mechanisms of immune evasion in Ewing sarcoma and osteosarcoma were highlighted. This

review summarizes the cutting-edge immunotherapy in Ewing Sarcoma and Osteosarcoma. However, the manuscript requires minor revisions.

1. I suggest that the authors use a table to illustrate Immunotherapy of Pediatric Bone Cancers.

2. I suggest that the author use different colors to distinguish the nucleus from the cytoplasm in Figure 2. 
